

**Impact of annual and seasonal precipitation and air**
**temperature on gross primary production in Mediterranean**
**ecosystems in Europe**
Svenja Bartsch[1], Bertrand Guenet[1], Christophe Boissard[1], Juliette Lathiere[1], Jean-
Yves Peterschmitt[1], Annemiek Stegehuis[1], Ilja-M Reiter[2], Thierry Gauquelin[3],
Virginie Baldy[3], Catherine Fernandez[3]
[1]Laboratoire des Sciences du Climat et de l'Environnement, LSCE/IPSL, CEA-
CNRS-OVSQ, Université Paris-Saclay, F-91191 Gif-sur-Yvette, France.
[2]Fédération de Recherche Ecosystèmes Continentaux et Risques Environnementaux
CNRS FR3098 ECCOREV, Domaine du Petit Arbois Avenue Louis Philibert,
Bâtiment du CEREGE - BP 80, 13545 Aix-en-Provence Cedex 04, France
[3]Aix Marseille Université, Avignon Université, CNRS, IRD, IMBE (Institut
Méditerranéen de Biodiversité et d'Ecologie marine et continental), 3 Place Victor
Hugo, 13331 Marseille, France
Corresponding author: Bertrand.Guenet@lsce.ipsl.fr



**Abstract**

22        Mediterranean ecosystems are significant carbon sinks but are also particularly

sensitive to climate change but the carbon dynamic in such ecosystem is still not fully
understood. An improved understanding of the drivers of the carbon fixation by plants
is needed to better predict how such ecosystems will respond to climate change. Here,
for the first time, a large dataset collected through the FLUXNET network is used to
estimate how the gross primary production (GPP) of different Mediterranean
ecosystems was affected by air temperature and precipitation between the years 1996
and 2013. We showed that annual precipitation was not a significant driver of annual
GPP. Our results also indicated that seasonal variations of air temperature
significantly affected seasonal variations of GPP but without major impact on inter
annual variations. Inter-annual variations of GPP seemed largely controlled by the
precipitation during early spring (March-April), making this period crucial for the
future of Mediterranean ecosystems. Finally, we also observed that the sensitivity of
GPP in Mediterranean ecosystems to climate drivers is not ecosystem type dependent.




### *1. Introduction*


Mediterranean land ecosystems are of particular interest for research
because their outstanding biodiversity is one of the most important after the
tropical regions (Cowling et al., 1996). This remarkable diversity is due to a
combination of biogeographical and environmental factors (soil types,
precipitation, temperature) but also the presence of human activities for millennia
(Lavorel et al., 1998; Rey Benayas and Scheiner, 2002). It has been hypothesized
that these ecosystems could be severely affected by global climate change in the
future, that includes modification of temperature and precipitation regime (Giorgi
and Lionello, 2008; Polade et al., 2014). $CO_2$ increase may also become an
important driver of species distribution within these regions (Keenan et al., 2011).
Mediterranean ecosystems supply numerous services to people including clean water,
flood protection and carbon sinks with a comparable amount of carbon uptake as
other European forests (Janssens et al., 2003). For instance, Vayreda et al., (2012) and
Pereira et al., (2007) observed a net ecosystem exchange (NEE) of 1.4 Mg C ha$^{-1}$ yr$^{-1}$
in a Spanish and Portuguese forest, and of 1.9 Mg C ha$^{-1}$ yr$^{-1}$ for a grassland in
Portugal, while an NEE of 2.7 Mg C ha-1 yr-1 was found for forest ecosystems from
the EUROFLUX network throughout Europe (Janssens et al., 2003)
Over the last decade considerable effort has been made to investigate the
effect of precipitation and air temperature on biomass production (Goerner et al.,
2009; Valladares et al., 2008). So far however, most of this research was carried out
using single site experiments (e.g. rain exclusion device (Limousin et al., 2009, 2010;
Martin-Stpaul et al., 2013)), or using only a few sites with a single ecosystem type
(Reichstein et al., 2002). Consequently, contrasting results are reported in the
literature. For instance, Reichstein *et al.*, (2002) observed a high sensitivity to drought



for three Mediterranean evergreen forests (two dominated by *Quercus Ilex* and one by
*Juniperus phoenicea)* whereas Grünzweig *et al.*, (2008) reported that another
Mediterannean species (*Quercus calliprinos*) was well adapted to drought. Sabaté *et*
*al.*, (2002) pointed out that Mediterranean oak forests (*Quercus Ilex*) were particularly
sensitive to summer drought whereas Allard *et al.*, (2008) observed an absence of
response to summer drought for another Mediterranean oak forest composed also by
*Quercus Ilex*. Moreover, Maselli, (2004) suggests that spring precipitation is the most
important factor controlling inter-annual variations of vegetation stress. To allow
broader conclusions, satellite monitoring on normalized difference vegetation index
has been performed (Maselli et al., 2014). However, the link between a vegetation
index and gross primary production (GPP) is not straightforward and there is a
substantial spread between different satellite products (Garrigues et al., 2008).

Up to now, we are not aware of any other study that has investigated the

impact of annual and seasonal precipitation and air temperature on the primary
production of Mediterranean ecosystems using a large collection of sites, under
different climatic conditions and different vegetation types. Model projections yet
indicate that the Mediterranean region will be strongly affected by future climate
change (Giorgi and Lionello, 2008; Guiot and Cramer, 2016; Polade et al., 2014). Our
approach might therefore be of great importance, providing general regional
information for modeling exercises, and enabling to improve future biomass
projections on a regional scale. A decrease in precipitation and an increase in
temperature, both associated with large spatial variability, are expected for the next
decades (Dubrovsky et al., 2014). This makes the Mediterranean region one of the
most vulnerable regions to climate change (Nissen et al., 2014) worldwide. In this





context, understanding the response of Mediterranean ecosystems to changes in
temperature and precipitation is of major importance.

The main goal of this study is to identify the impact of annual and seasonal

precipitation (PPT) and air temperature (T) on GPP throughout the European
Mediterranean region, based on a multi-sites analysis. To do so, we applied ANOVAs
and linear mixed effect models to the GPP obtained from 23 different FLUXNET
sites in the European Mediterranean area, representing 5 different ecosystem types.

*2. Material and Methods*
**2.1. Data set & data organization**

We used the FLUXNET database (http://www.fluxdata.org), which contains

flux measurements ($CO_2$, water, etc.) based on the eddy covariance method
(Baldocchi et al., 2001) and meteorological measurements at a high temporal
resolution (up to 30-min intervals). The database covers more than 500 registered
sites worldwide and is partly freely available under a Fair-Use policy. All data
provided by the international FLUXNET network are processed according to
standardized formats and data processing protocols (Moffat et al., 2007; Papale et al.,
2006; Reichstein et al., 2005).

In this study we used level 4 data (L4, daily time-steps) of GPP, PPT and T

from the La Thuile collection (see also http://www.fluxdata.org). We selected sites
that are located within the Mediterranean region with the following vegetation types:
shrubs (S), grasslands (G), deciduous broadleaf trees (DBT), evergreen needle trees
(ENT) and evergreen broadleaf trees (EBT). We only focused on the European region.
From the site-year files we calculated the annual mean and sum values of GPP, PPT
and T. We also included the corresponding vegetation types. To be able to investigate



the impact of different seasons in a rather precise way, we split the year into six parts
using a bi-monthly time step (January & February (JF), March & April (MA), May &
June (MJ), July & August (JA), September & October (SO), November & December
(ND)) (cf. Tab. 2 subset S0-S6).

We only considered the site-year files where at least 90% of the needed data

per year or bi-monthly time step were available. This selection process resulted in 23
sites in three different countries (France, Italy, Spain) as presented in table 1.

**2.2. Statistical methods**

The statistical analyses were performed using RStudio (version 0.99.473,

2009-2015 RStudio). The impact of annual and seasonal PPT, T and the vegetation
type on annual and seasonal GPP was investigated by employing a linear mixed effect
model ANOVA. The sites were used as random effect, which enabled us to take
potential site-dependency effects into account. Because of non-normality, data were
rank-transformed before analysis, as previously done by e.g. Guenet *et al.* (2014).

We tested seven different subsets (Tab. 2, S0-S6). We first investigated if the

annual mean PPT, T and/or vegetation type significantly affected the annual mean
GPP (Tab. 2, S0, case A). Then, we analyzed the annual GPP using bi-monthly mean
instead of annual mean PPT and T values (Tab. 2, S1-S6, case A). Note that we
investigated the impact of PPT and T on the average annual GPP of the subsequent
year rather than on the actual year for the ND subset, because in this time of the year
the actual climatic factors hardly control the total growing strength of the actual year
(Tab. 2). In a next step all tests were repeated using the total annual and bi-monthly
sum, instead of mean values for GPP and PPT (Tab. 2, S0-S6, case B). As we applied
several hypotheses on one single data set, we faced the problem of multiple



comparisons minimizing the probability of receiving a Type I error. Accordingly, we
corrected the original significance level (p = 0.05) by applying the Holm-Bonferroni
method (Holm, 1979). In a last step, we investigated if the PPT and T of specific
seasons (bi-monthly time periods) significantly affected the GPP of the corresponding
seasons (Tab. 2, S1-S6, case C). In the latter case, applying the Holm-Bonferroni
method was not necessary as we used an independent data set for every season and
subset.
To interactively explore which predictors provided a good fit, we applied a
stepwise regression in all cases, which conducts an automatic stepwise model
selection by the AIC (Akaike information criterion).

### *3. Results*
**3.1. Inter-annual GPP variability**
Over the selected sites, the vegetation faced a typical Mediterranean climate,
with usually hot and dry summers as well as mostly mild and moist winters (Fig. 1). T
ranged from -0.1 to 28.4°C (Fig. 1C, bi-monthly averages) and the seasonal PPT from
0 to 11.4 mm d$^{-1}$ (Fig. 1D, bi-monthly averages). GPP values for shrubs were lowest
but show the highest variability across the different sites (Fig. 1A). For trees (ENT,
DBT, EBT) the GPP values were rather similar to each other. For grassland only two
suitable sites were available (Tab. 1, Fig. 1A).
Surprisingly no significant correlation was found between annual GPP and
annual T or annual PPT across sites and years. A general trend over the vegetation
types was observed but this was not significant according to the Holm-Bonferroni
corrected threshold (Tab. 3). Furthermore, the relationships between the applied
climatic factors and vegetation types were never found to be significant (Tab. 3). We



did not obtain a clearly relationship between annual and seasonal PPT and annual
GPP, or between annual and seasonal T and annual GPP by simply applying linear
regression models (Fig. 2-5). Nevertheless, annual GPP averages could be explained
(significant p-values) by both precipitation during early spring (MA) and air
temperature during early winter (ND), when using bi-monthly averages or the sum as
explaining variables in the linear mixed effect model (Tab. 3, Fig. 2 & 4).
Furthermore, we found that the annual GPP was not significantly affected by climate
conditions in summer (MJ & JA), even though this time period is the hottest and the
driest for all sites (Fig. 1). Finally, we noticed that none of the interactions between
the explaining variables, vegetation type, T (bi-monthly & annual) and PPT (bi-
monthly & annual), significantly impacted the annual average of GPP (Tab. 3).

**3.2. Intra-annual GPP variability**
We observed that GPP was low at the beginning of the year (JF) and increased
till MJ (highest median value 6.8 gC m$^{-2}$ d$^{-1}$), when looking at the bi-monthly
distribution of GPP (Fig. 1B). During the summer GPP slowly decreased until the
lowest median value in ND (2.2 gC m$^{-2}$ d$^{-1}$). The highest variability in GPP was
observed in JA and dominated by broadleaf trees.
During all time periods the bi-monthly average T significantly affected the bi-
monthly average GPP (Tab. 3). In general we observed a positive relationship
between seasonal T and seasonal GPP. The only exception occurred in JA, when the
increasing air temperature caused a decrease in GPP. From May to August the bi-
monthly average GPP was additionally significantly affected by the bi-monthly
average of PPT. Finally, as observed for the inter-annual variations of GPP, none of





the interactions between the seasonal PPT, seasonal T and vegetation types was
significantly correlated to the bi-monthly average of GPP.

*4. Discussion*
Interestingly, neither the annual T nor the annual PPT was found to be a major
control on annual biomass production in the Mediterranean region. This underlines
the importance of applying seasonal (or intra-annual) approaches rather than mere
inter-annual studies when investigating potential effects on biomass production within
the Mediterranean region. This result is in contradiction with Jongen *et al.*, (2011)
who observed a positive correlation between annual precipitation and GPP for a
Portuguese grassland. However, we only had data for two grassland sites, hence the
observed trends were largely controlled by forests and shrublands. Our results are
however in accordance with Allard *et al.* (2008) who observed that the seasonal
averages of precipitation, more than the annual average, were important drivers for
GPP.
The rainfall during the early spring months (MA) had an important impact on
annual GPP. PPT over the other time periods however, did not significantly affect
annual GPP. During MA, when the growing season starts, the rainfall (Fig. 1D) is
high enough to support vegetation growth, whereas the air temperature is not yet too
high to reduce C fixation (Fig. 1C). Hence, early spring does not only provide good
growing conditions, it can also control the soil moisture conditions before extremely
dry and hot summer months (see Fig. 1B & C). The highest GPP values as well as the
highest GPP variability were observed in the summer months (MJ, JA; Fig.1B). MA
can thus be seen as a decisive time period in the year in controlling the annual
biomass production. Allard *et al.*, (2008) concluded that a decrease of precipitation in



April-June would have a large effect on annual net ecosystem production (NEP),
whereas the impact of decreasing precipitation in July-September on NEP would be
minimal. Our results are also consistent with the work of Maselli (2004), who
reported that normalized difference vegetation index was mainly affected by spring
precipitation. These results highlight the importance of the distribution of
precipitation within a year, rather than the annual precipitation sum.

If climate change would affect early spring precipitation, the effect on GPP

might be highly significant, whereas if precipitation would change during the other
periods, the effect on GPP of Mediterranean ecosystems might be limited. Polade et
al., (2014) showed that Mediterranean climate regions would face a dryer climate in
the future. Using a multi-scenario and a multi-model ensemble, Goubanova & Li
(2007) suggested that the precipitation over the Mediterranean region will be reduced
throughout the 21$^{st}$ century during spring, summer and autumn. Moreover,
Toggweiler & Key (2001) showed that the intensity of precipitation will mostly be
reduced from February to May, using downscaling statistics. Taken together, these
results indicate that the effect of climate change on the GPP of Mediterranean
ecosystems might be very important, leading to a reduction of total carbon storage
from those ecosystems. It furthermore suggests that the effects of climate change
should be estimated using mechanistic models with a fine time resolution or with
statistical models based on precipitation distribution at a seasonal time scale.
Interestingly, the winter T also controls the annual GPP (Tab. 3). Lowest GPP values
were associated to lower air temperatures. Most Mediterranean vegetation is well
adapted to heat and water stress, but not all species might be able to survive low
winter temperatures (Aranda et al., 2005; Ferrio et al., 2003; Karavatas and Manetas,
1999; Larcher, 2000; Llorens et al., 2003). These can induce photoinhibition that



cannot be reversed during periods with more favorable temperatures (Camarero et al.,
2012; Ogaya et al., 2011). Furthermore, below-zero temperatures may induce
freezing-induced embolism, which can only be partly restored (Cochard et al., 2001;
Nardini et al., 2000). These factors can predispose trees to drought and heat stress that
are often occurring during summer in the Mediterranean region (Peguero-Pina et al.,
2011). Bansal *et al.*, (2015), and Sohn et al., (2012) found that winter conditions are
most likely more decisive for plant growth than summer aridity in some parts of the
Mediterranean region. However, they may also be partly favorable for reducing
summer stress, as they may influence the genetic variation in drought-resistance
because traits of drought- and freezing-resistance can be co-occurring (Bansal et al.,
2015; Blödner et al., 2005; Gimeno et al., 2009) However, future climate projections
suggest an increase of air temperature over the Mediterranean regions (Goubanova
and Li, 2007), reducing the impact of cold winter temperatures on GPP. For air
temperature and precipitation it is interesting to note that interactions with vegetation
types were never significant, suggesting that the observed responses are not dependent
on the vegetation types.

Our results also illustrate that summer PPT as well as summer T (for both MJ

& JA) significantly influenced summer GPP (see Tab.3C). In particular, increasing JA
T caused a decrease in GPP. Allard *et al.*, (2008) presented the Mediterranean region
as an area characterized by a long growing season that is often interrupted during late
summer, when water stress is getting too high (see also Reichstein *et al.*, (2002)). The
authors suggested that under these extreme drought conditions, GPP and ecosystem
respiration ($R_{eco}$) are partly decoupled, most likely due to stomatal closure. Our results
slightly support this finding. We observed a negative correlation between T and GPP
during JA (Fig. 5, second line, right-side plot). Hence, if T reaches a certain threshold,



GPP starts to decrease. In general, seasonal T seems to control the equivalent seasonal
GPP, which indicates that seasonal T, and not the PPT as expected, generally has a
major effect on seasonal GPP in the Mediterranean regions. However, the seasonal
effect of T on GPP during the same season seems not to directly affect the annual
GPP.
**5. Conclusions**
In this study we investigated the response of the GPP of Mediterranean
ecosystems to different climatic variables. We used the largest collection of sites over
the Mediterranean region that has been considered so far. We showed that seasonal
variations of T significantly impacted seasonal variations of GPP, but without a major
impact on the variation of inter annual GPP. Our results suggest that variations of
inter annual GPP are largely controlled by early spring precipitation, making this
period crucial for the future of Mediterranean ecosystems. Interestingly, we did not
observe an effect of vegetation type, indicating that the response of GPP of
Mediterranean ecosystems to climate drivers is not vegetation-type dependent.
Unfortunately, the studied sites were only located in Europe. To broaden our
conclusions more data would be needed from other non-European parts of the
Mediterranean region. Nevertheless, we showed that in the future, the reduction of
spring precipitation will have a major impact on carbon storage of many different
Mediterranean ecosystems.




**Acknowledgments:**
This work was supported by the French National Agency for Research (ANR-
12-BSV7-0016-01, SECPRIME2).

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





**List of Figures**
**Fig. 1**: Boxplots showing A) the general GPP distribution of the different vegetation
types (ENT = evergreen needle trees; DBT = deciduous broadleaf trees; EBT = ever-
green broadleaf trees; G = grasslands; S = Shrubs; numbers in the brackets indicating
the numbers of sites per vegetation type) and (B) the GPP distribution C) the air
temperatures (T) and D) the precipitation distribution observed during the different bi-
monthly time periods (JF = January & February, MA = March & April, MJ = May &
June, JA = July & August, SO = September & October, ND = November &
December).

**Fig. 2**: Seasonal mean PPT versus the annual mean GPP for the different vegetation
types and over the different bi-monthly time periods. See Fig. 1 for the abbreviations.
A simple trend line & R-squared value (including all vegetation types) was added to
those plots where a significant p-value was obtained during our statistical tests (see
Tab. 3).

**Fig. 3**: Seasonal mean PPT versus the seasonal mean GPP for the different vegetation
types and over the different bi-monthly periods. See Fig. 1 for the abbreviations.
Trend lines & R-squared values (including all vegetation types) were added to those
plots where a significant p-value was obtained during our statistical tests (see Tab. 3).

**Fig. 4**: Seasonal mean T versus the annual mean GPP for the different vegetation
types and over the different bi-monthly time periods. See Fig. 1 for the abbreviations.
A simple trend line & R-squared value (including all vegetation types) was added to





those plots where a significant p-value was obtained during our statistical tests (see
Tab. 3).

**Fig. 5**: Seasonal mean T versus the seasonal mean GPP for the different vegetation
types and over the different bi-monthly time periods. Trend lines & R-squared values
(including all vegetation types) were added to those plots where a significant p-value
was obtained during our statistical tests (see Tab. 3). See Fig. 1 for the abbreviations.







**Figures**

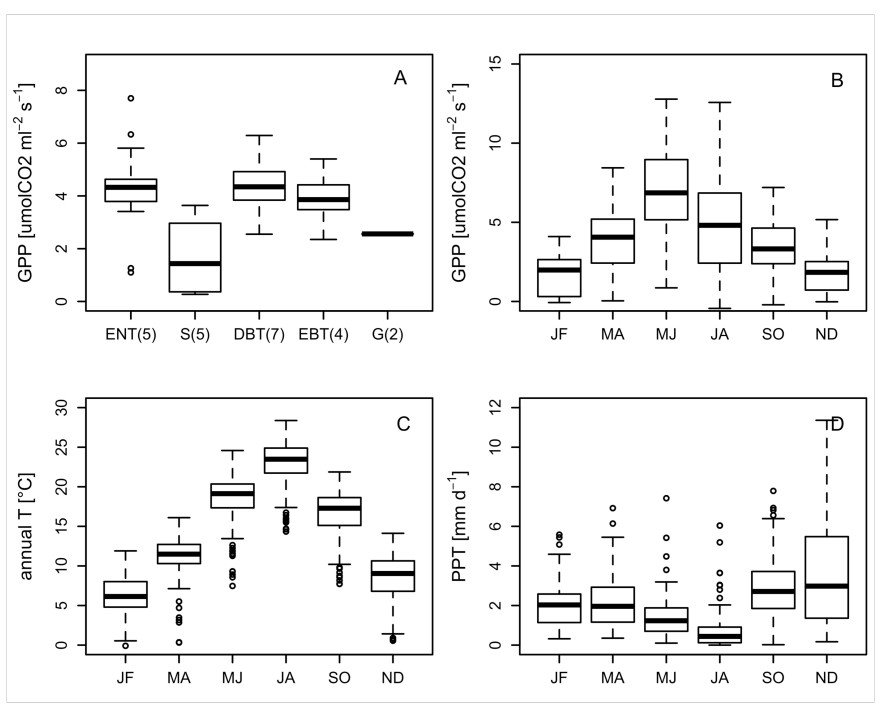


521                                    **Figure 1**





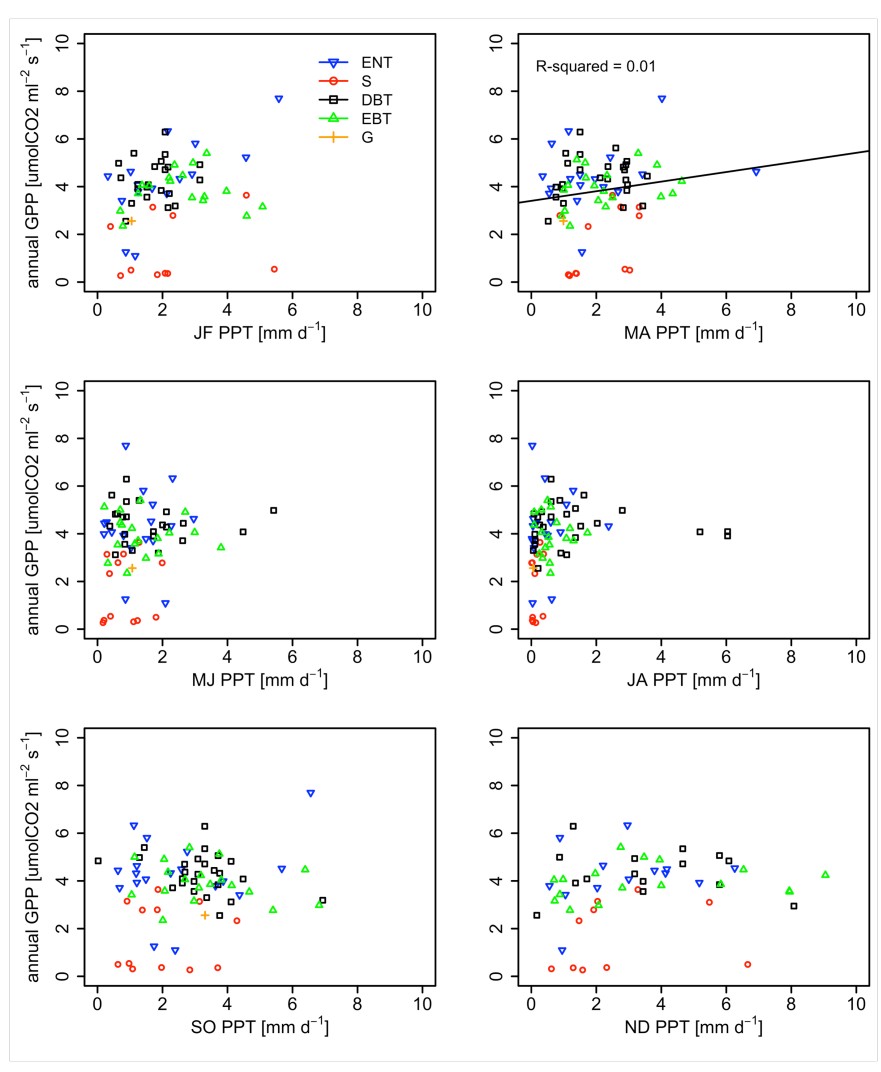


523                                        **Figure 2**





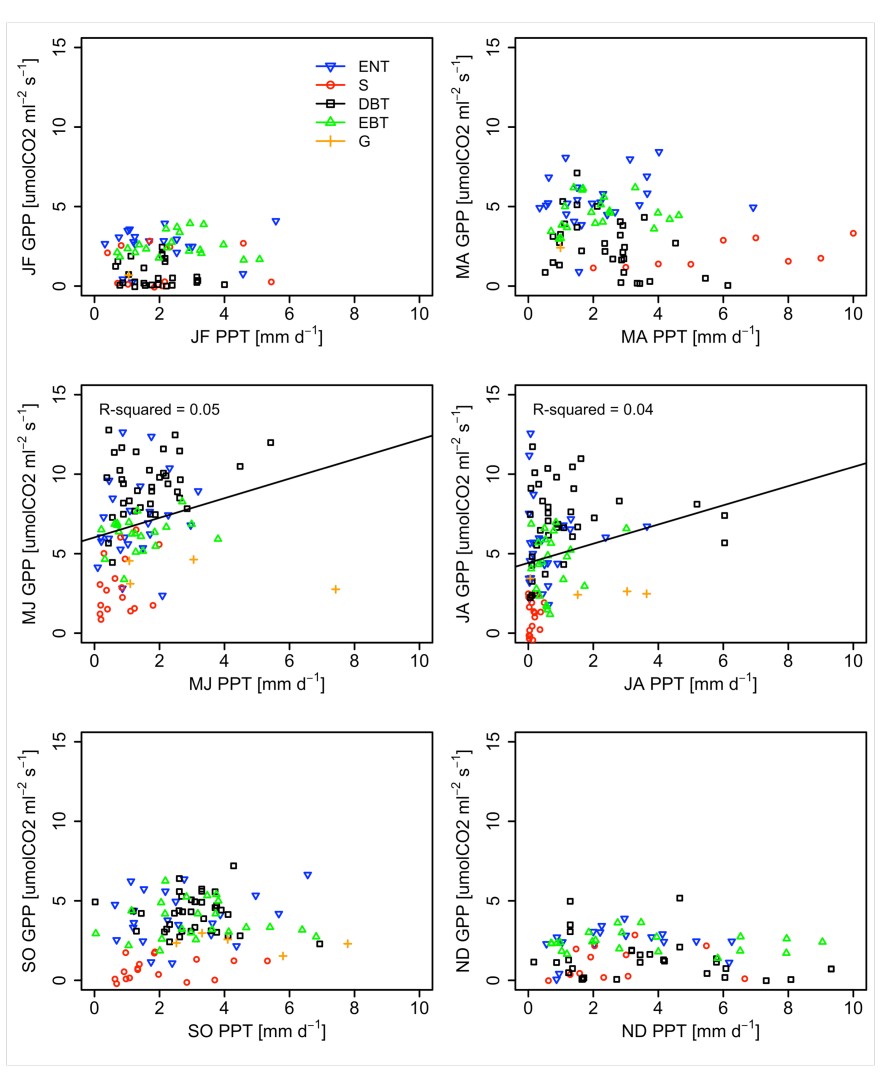


525                                **Figure 3**





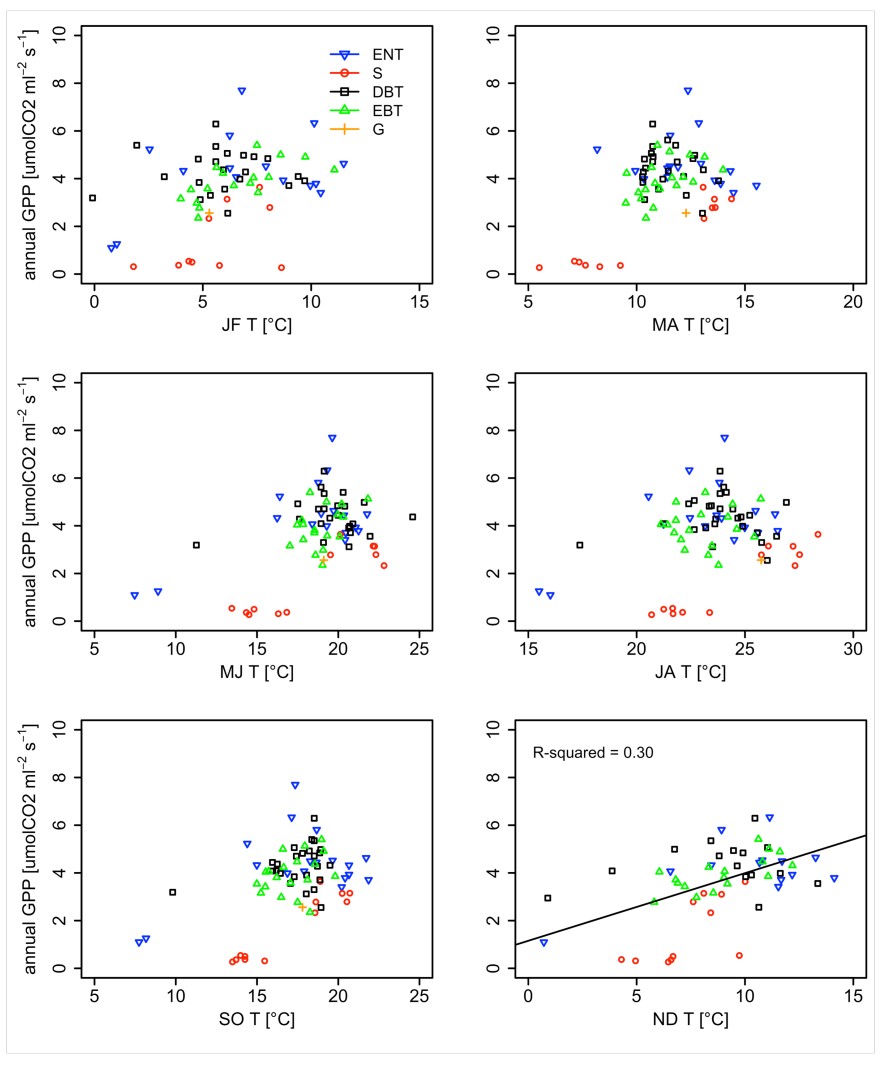


527                                    **Figure 4**





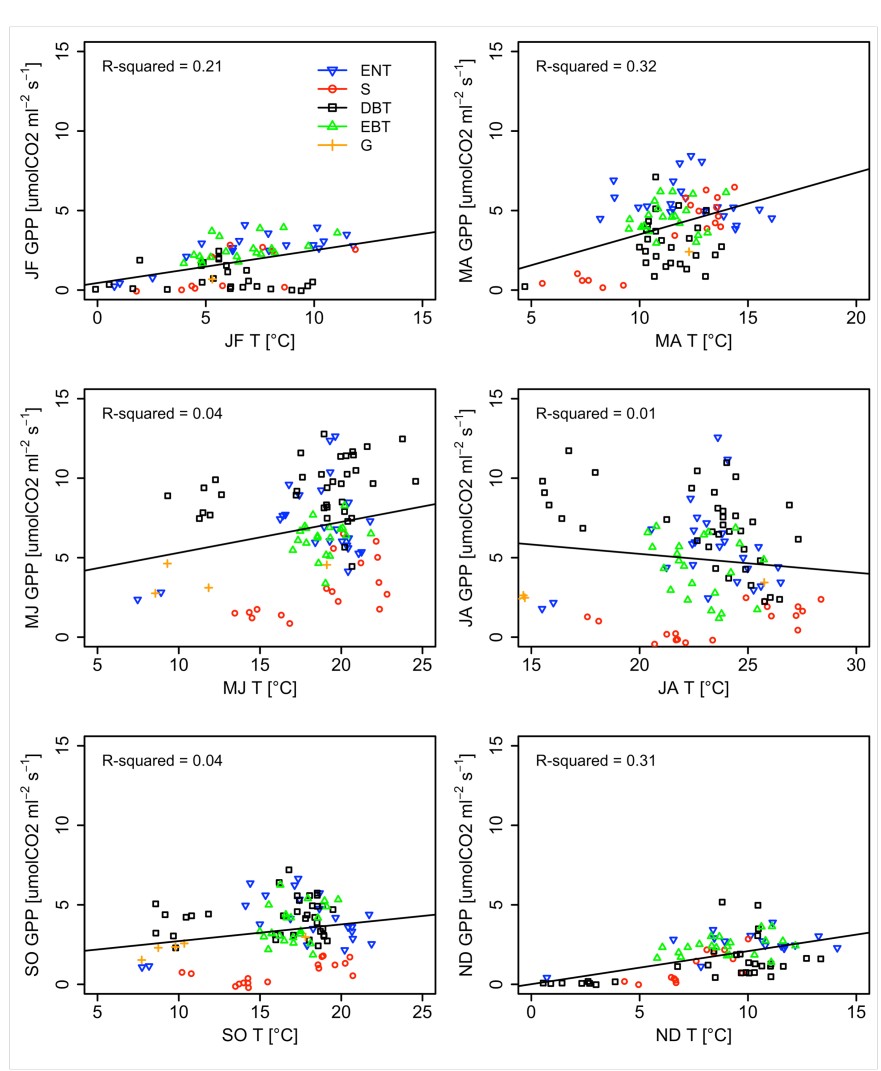


**Figure 5**







**Tables**

**Tab. 1**: Site description

| Nr. | SITE ID | SITE NAME | COUNTRY | COORDINATES (Lat., Long.) | VEGETATION |
|---|---|---|---|---|---|
| 1 | ESES1 | El Saler | Spain | 39.3460, -0.3188 | evergreen needleleaf trees |
| 2 | ESLgS | Laguna Seca | Spain | 37.0979, -2.9658 | evergreen needleleaf trees |
| 3 | ESLJu | Llano de los Juanes | Spain | 36.9266, -2.7521 | shrubs |
| 4 | ESLMa | Las Majadas del Tietar | Spain | 39.9415, -5.7734 | shrubs |
| 5 | ESLn1 | Lanjaron-Non intervention | Spain | 36.9721, -3.4739 | shrubs |
| 6 | ESLn2 | Lanjaron-Salvage logging | Spain | 36.9695, -3.4758 | shrubs |
| 7 | ESVDA | Vall d'Alinya | Spain | 42.1522, 1.4485 | grasslands |
| 8 | FRFBn | Font-Blanche | France | 43.2408, 5.6792 | evergreen needleleaf trees |
| 9 | FRPue | Puechabon | France | 43.7414, 3.5958 | evergreen broadleaf trees |
| 10 | ITBon | Bonis | Italy | 39.4778, 16.5347 | evergreen needleleaf trees |
| 11 | ITCA1 | Castel d'Asso1 | Italy | 42.3804, 12.0266 | deciduous broadleaf trees |
| 12 | ITCA3 | Castel d'Asso2 | Italy | 42.3772, 12.0260 | grasslands |
| 13 | ITCol | Collelongo-Selva Piana | Italy | 41.8494, 13.5881 | deciduous broadleaf trees |
| 14 | ITCpz | Castelporziano | Italy | 41.7052, 12.3761 | evergreen broadleaf trees |
| 15 | ITLec | Lecceto | Italy | 43.3036, 11.2698 | evergreen broadleaf trees |
| 16 | ITNon | Nonantola | Italy | 44.6902, 11.0911 | deciduous broadleaf trees |
| 17 | ITPia | Island of Pianosa | Italy | 42.5839, 10.0784 | shrubs |
| 18 | ITRo1 | Roccarespampani1 | Italy | 42.4081, 11.9300 | deciduous broadleaf trees |
| 19 | ITRo2 | Roccarespampani2 | Italy | 42.3903, 11.9209 | deciduous broadleaf trees |
| 20 | ITSRo | San Rossore | Italy | 43.7279, 10.2844 | evergreen needleleaf trees |
| 21 | ITTo1 | Tolfa wet | Italy | 42.1897, 11.9216 | deciduous broadleaf trees |
| 22 | ITTo2 | Tolfa dry | Italy | 42.1897, 11.9216 | deciduous broadleaf trees |
| 23 | ITTol | Tolfa | Italy | 42.1897, 11.9216 | evergreen broadleaf trees |

**Tab. 2**: Annual and bi-monthly subsets (PPT = precipitation; T = air temperature). Note that for the early winter (ND) subset we studied the impact of the PPT and T on the average annual GPP of the subsequent year rather than on the actual year.

| ID | Subset description |
|---|---|
| S0 | mean annual (Jan.-Dec.) PPT, T & vegetation |
| S1 | mean JF (Jan. & Feb.) PPT, T & vegetation |
| S2 | mean MA (Mar. & Apr.) PPT, T & vegetation |
| S3 | mean MJ (May & Jun.) PPT, T & vegetation |
| S4 | mean JA (Jul. & Aug.) PPT, T & vegetation |
| S5 | mean SO (Sept. & Oct.) PPT, T & vegetation |
| S6 | mean ND (Nov. & Dec.) PPT, T & vegetation |

(A) Impact on average annual GPP
(B) Impact on total annual GPP (sum)
(C) Impact on average seasonal GPP



**Tab. 3**: Results of the statistical analysis. The numbers represent significant p-values ($p < 0.05$) whereas an 'o' represents no significance ($p > 0.05$). The used significance levels are given in brackets. Green numbers are representing p-values that are still significant after the Holm-Bonferroni correction. Red numbers indicate the p-values that lost their significance after the Holm-Bonferroni correction. Black numbers represent the applied significance level without using the Holm-Bonferroni correction (Seasonal Approach).

| | annual (S0) | seasonal (S2-S7) | | | | | |
| --- | --- | --- | --- | --- | --- | --- | --- |
| | | Jan & Feb (S2) | Mar & Apr (S3) | May & Jun (S4) | Jul & Aug (S5) | Sep & Oct (S6) | Nov & Dec (S1) |
| Nr. of sites | 19 | 19 | 19 | 18 | 19 | 19 | 15 |
| Nr. of years | 76 | 65 | 74 | 70 | 74 | 75 | 56 |
| $R^2$ | 0.64 | 0.70 | 0.71 | 0.75 | 0.69 | 0.70 | 0.70 |
| **(A) averages** | | | | | | | |
| PPT | o | o | 0.008 (p < 0.050) | o | o | o | o |
| T | o | 0.022 (p < 0.008) | 0.047 (p < 0.006) | o | o | o | 0.011 (p < 0.025) |
| vegetation | 0.020 (p < 0.010) | 0.027 (p < 0.007) | 0.016 (p < 0.017) | o | o | 0.030 (p < 0.006) | 0.019 (p < 0.013) |
| PPT:T | o | o | o | o | o | o | o |
| PPT:vegetation | o | o | o | o | o | o | o |
| T:vegetation | o | o | o | o | o | o | o |
| $R^2$ | 0.63 | 0.70 | 0.70 | 0.73 | 0.68 | 0.69 | 0.96 |
| **(B) sums** | | | | | | | |
| PPT | o | o | 0.012 (p < 0.025) | o | o | o | o |
| T | o | 0.020 (p < 0.010) | 0.030 (p < 0.006) | o | o | o | 0.008 (p < 0.050) |
| vegetation | 0.016 (p < 0.013) | 0.027 (p < 0.007) | 0.014 (p < 0.017) | o | o | 0.025 (p < 0.008) | 0.027 (p < 0.006) |
| PPT:T | o | o | o | o | o | o | o |
| PPT:vegetation | o | o | o | o | o | o | o |
| T:vegetation | o | o | o | o | o | o | o |
| $R^2$ | - | 0.84 | 0.67 | 0.78 | 0.73 | 0.63 | 0.78 |
| **(C) seasonal approach** | | | | | | | |
| PPT | - | o | o | 0.003 (p < 0.050) | 0.003 (p < 0.050) | o | o |
| T | - | 0.001 (p < 0.050) | 0.009 (p < 0.050) | 0.005 (p < 0.050) | 0.005 (p < 0.050) | 0.041 (p < 0.050) | 0.003 (p < 0.050) |
| vegetation | - | o | o | o | o | 0.009 (p < 0.050) | o |
| PPT:T | - | o | o | o | o | o | o |
| PPT:vegetation | - | o | o | o | o | o | o |
| T:vegetation | - | o | o | o | o | o | o |