# Peer review of "Impact of annual and seasonal precipitation and air 1 temperature on gross primary production in Mediterranean 2 ecosystems in Europe 3 4 Svenja Bartsch1, Bertrand Guenet1, Christophe Boissard1, Juliette Lathiere1, Jean-5 Yves Peterschmitt1, Ann"

_Biogeosciences, 2016_

## Referee Comment (RC1) · Anonymous Referee #1 · 12 Dec 2016

Please see 3 attached images with screen captures of review text.

**General Comment:**

The authors have conducted a statistical analysis of FLUXNET database products to examine the dependence of gross primary production (GPP) on environmental factors in 23 Mediterranean ecosystems. In my opinion, this represents a misuse of the FLUXNET database, and the scientific approach is not valid as a result. The reasons for this opinion are as follows:

- The main variable under consideration (GPP) is not directly measured, but is the result of an undocumented (in this manuscript) flux decomposition technique whose validity is highly questionable in water-limited ecosystems such as those considered here. Flux towers make direct measurements of net ecosystem exchange (NEE), often considered to be the difference between ecosystem respiration ($R_{eco}$) and GPP:

$$NEE = R_{eco} - GPP \qquad (1)$$

Traditionally, the means of decomposing NEE is to
  - Assume at night (GPP=0) that $R_{eco}$ is directly measured;
  - Assume that $R_{eco}$ depends only on the temperature (e.g., $Q_{10}$);
  - Model daytime $R_{eco}$ based on the temperature; and
  - Derive GPP from direct measurements of NEE and modelled $R_{eco}$.

Such a flux decomposition algorithm tends to perform poorly in water-limited ecosystems, where $R_{eco}$ is often suppressed by drought conditions (and drought may be defined in terms of limited soil water, or excessive air dryness, or both). We have found this to be the case in the Mediterranean, rather generally with the exception of wet winter months. For this reason and until a valid flux partitioning scheme can be derived for Mediterranean ecosystems, the FLUXNET data product GPP should be considered highly dubious in dryland ecosystems, and should not be blindly supposed to be an accurate assessment of plant activity.

**Fig. 1.**

- A second consideration is that equation (1) above fails in many semi-arid ecosystems, and this has been demonstrated amply at el Llano de los Juanes (site number 3 in Table 1). As has been documented by local investigators in numerous publications, NEE is not directly related to GPP and $R_{eco}$ on half-hour timescales, but is buffered by the accumulation of $CO_2$ in the underground environment, and dominated by ventilation when such stored $CO_2$ is flushed by winds or pressure fluctuations . Spanish researchers have worked to document the existence of these additional flux components (1), verify their effects via independent underground measurements of $CO_2$ stocks (2), confirm their relevance as worthy of accounting in the annual balance, relative to GPP and $R_{eco}$ (3), and propose flux decomposition methods to model their dependence on environmental variables (4). The neglect of all of this work by the authors of the manuscript under review leads directly to my next criticism.
- The authors appear to have greatly overlooked the knowledge of local investigators regarding their sites. Most of the sites in Table 1 are towers managed by experienced groups with publications regarding these very sites. Generally, the authors of the submitted manuscript have not sited the relevant papers which include site descriptions, and even previous descriptions of ecosystem functioning that would make some of the authors' observations seem less "surprising". Prior to publishing

**Fig. 2.**

analyses of data from these sites, I suggest that proper conduct in science would be to consult the local investigators, perhaps invite them to collaborate, and at least read/cite their papers where appropriate.

For all of the above reasons, and based on the guidelines set forth by Biogeosciences, I consider that the scientific approach applied in this manuscript is not valid, that the results are not "discussed in an appropriate and balanced way (consideration of related work, including appropriate references)", and therefore conclude that the manuscript should be rejected.

References Sited

1. Serrano-Ortiz et al., 2010, *Agricultural and Forest Meteorology*, **150** (3), 321-329.
2. Sánchez-Cañete et al., 2011, *Geophysical Research Letters*, doi:10.1029/2011GL047077.
3. Serrano-Ortiz et al., 2009, *Journal of Geophysical Research*, doi:10.1029/2009JG000983.
4. Pérez-Priego et al., 2013, *Agricultural and Forest Meteorology*, **180**, 194 – 202.

**Fig. 3.**

---

## Referee Comment (RC2) · Anonymous Referee #2 · 18 Dec 2016

General comment Overall, the topic presented by the authors would be fitting to the scope of BGS and, also, is of interest in the "response to climate change" of ecosystems, of relevance for process understanding.

However, the papers suffers of major drawbacks, the most important is the not completely correct consideration of the selected sites with additionally a lack of information/data on sites that would allow the readers (and referees) to evaluate the sites in the perspectives of what the authors want to affirm.

When assessing the impact of T/PPT on GPP or another flux parameters, a reasearcher should be sure that climate (T/PPT) is the driver that can be really considered for such an analysis.

[Figure]

For example, in the case of the selected sites: - site 6. It is subject to salvage logging, after a fire or similar (no reference is given). Here "management" (salvage logging) could be expected to seriously influence production (GPP, NPP) and respiration, with potential "decoupling" from climate influence. - site 11. This site is a short rotation forestry stand where underground irrigation and fertilisation was applied. Also in this case, GPP may be decoupled from PPT, particularly in summer - site 21 and 22: it is the same site, with two stands under a treatment with increased water supply (wet) and reduced water supply (dry) by using water interception gauges in the dry treatment and relocation of the water in the wet one. Also in this case, PPT is not the proper variable to be possibly considered, unless information on effective water supply has been used in the analysis. Furthermore, those two sites are evergreen broadleaf trees, as site 23 (which is the same site with natural water input).

As the authors grouped the sites according to vegetation, an incorrect assignment of a site to the current vegetation hamper the analysis, averages, box plots and the subsequent analysis.

Apart from this fundamental comment, table 1 with site description does not provide mean climate data, elevation (as a minimum), main species, to allow the reader/referee to have a clearer view of what are the mean conditions at the sites. Also, it would be interesting to know the number of site/years used for each site. A reference to published work from those sites is needed (for correctness but also to have a better picture of what the site is)

The figures are lacking of a "symbol legend", it is not clear which symbol represents which system/site.

Although the topic is of interest, the current quality of the paper and the drawbacks illustrated above suggests me to reject the paper.
* * *

---

## Author Comment (AC1) · 7 Feb 2017

**General Answer to the Reviewer: We thank the anonymous reviewers for reading and reviewing our manuscript. We agree with both reviewers that our site selection could benefit from improvement and that we mainly concentrated on the vegetation types, not respecting the applied management or treatment of each site. We also agree that more information has to be provided concerning the selected sites. Correspondingly, we corrected our site selection by excluding sites where the management is difficult to reconcile with our statistical analysis. As given in table 1 (Tab.1, see below) we ended up having 16 sites left (instead of the original 23), representing four different vegetation types (evergreen needleleaf trees; evergreen broadleaf trees; deciduous**

[Figure]

broadleaf trees & shrubs) and three countries (Spain; France & Italy). In addition we included missing information such as the total elevation; climate information (KGCC); the number of years of observations included per site, as well as publications that are relevant to the sites' description (this column will be completed) in table 1. Based on this updated site selection, we will re-run all statistical analysis on the data.

**Reply to the comments of reviewer 1:**

**We understand the first concern of reviewer 1. Nevertheless there already have been publications (see below) using GPP of water-limited systems out of the FLUXNET data sets, e.g.:**

Ross, I., Misson, L., Rambal, S., Arneth, A., Scott, R. L., Carrara, A., Cescatti, A., and Genesio, L.: How do variations in the temporal distribution of rainfall events affect ecosystem fluxes in seasonally water limited Northern Hemisphere shrublands and forests?, Biogeosciences, 9, 1007–1024, doi:10.5194/bg-9-1007-2012, 2012.

Quotation from Ross et al. (2012): "Flux tower data allow direct quantification of NEP and its decomposition into GPP and RE (Reichstein et al., 2005) and make it possible to analyze relationships between ecosystem fluxes and rainfall characteristics across ecosystem types and sites in a robust way."

We think however, that it is still a quite interesting point to consider in our manuscript. Therefore, we have decided to run all our statistical analysis also for NEE (as a 'real' measurement). We will compare the results on NEE and GPP to see if it will underline effects such as e.g. additionally flux components as described in the literature presented by reviewer 1. Finally, our discussion on this particular point was rather poor and we will discuss these aspects more carefully in the new version of the manuscript.

**For the second point, we want to emphasize that we included 'site' as a random factor into our statistical analysis. Hence, if there is a site-specific effect it will be considered in our analysis. Nevertheless, the discussion section will be completed in the next**

version of the manuscript.

**For the third point, as given in the new table 1 we added several columns including relevant publications as well as some additional site information. We also want to apologize at this point that we did not yet acknowledge the FLUXNET network and its tremendous achievements. We highly appreciate this work and the opportunity to use these very well organized data sets. The FLUXNET network will be properly acknowledged in the next version.**

**Finally, we do not fully agree that our results are generally discussed in an unbalanced way. However, we are planning (as mentioned previously) to add several aspects pointed out in the reviewers' comments, such as potential accumulation of $CO_2$ in the underground, to our discussion part.**
* * *
Tab.1: Site description (NEW SITE SELECTION). Further site information is available at: https://fluxnet.ornl.gov/.

| Nr. | SITE ID | SITE NAME | COUNTRY | COORDINATES (Lat., Long.) | VEGETATION | ELEVATION | KGCC *1 | YEARS *2 | REFERENCES |
|---|---|---|---|---|---|---|---|---|---|
| 1 | ES-ES1 | El Saler | Spain | 39.3460, -0.3188 | evergreen needleleaf trees | 10 m | Csa | 1999 - 2006 | Sanz et al. (2004) |
| 2 | ES-LgS | Laguna Seca | Spain | 37.0979, -2.9658 | shrubs | 2267 m | Csa | 2007 - 2008 | - |
| 3 | ES-Llu | Llano de los Juanes | Spain | 36.9266, -2.7521 | shrubs | 1600 m | Csa | 2005 - 2011 | Serrano-Ortiz et al. (2007) |
| 4 | ES-Ln1 | Lanjaron-Non intervention | Spain | 36.9721, -3.4739 | shrubs | 2301 m | Csa | 2009 | - |
| 5 | FR-FBn | Font-Blanche | France | 43.2408, 5.6792 | evergreen needleleaf trees | 436 m | Csa | 2009 - 2011 | - |
| 6 | FR-Pue | Puechabon | France | 43.7414, 3.5958 | evergreen broadleaf trees | 270 m | Csa | 2001 - 2011 | Rambal et al. (2004) |
| 7 | IT-Bon | Bonis | Italy | 39.4778, 16.5347 | evergreen needleleaf trees | 1170 m | Csa | 2005 - 2009 | - |
| 8 | IT-CA3 | Castel d'Asso3 | Italy | 42.3772, 12.0222 | deciduous broadleaf trees | 197 m | Csa | 2012 | - |
| 9 | IT-Cpz | Castelporziano | Italy | 41.7052, 12.3761 | evergreen broadleaf trees | 68 m | Csa | 1997, 2000 - 2008 | Garbulsky et al. (2008) |
| 10 | IT-Lec | Lecceto | Italy | 43.3036, 11.2698 | evergreen broadleaf trees | 314 m | Cfa | 2005 - 2009 | Chiesi et al. (2011) |
| 11 | IT-Non | Nonantola | Italy | 44.6902, 11.0911 | deciduous broadleaf trees | 20 m | Cfa | 2001 - 2003, 2006 - 2008 | Reichstein et al. (2003) |
| 12 | IT-Pia | Island of Pianosa | Italy | 42.5839, 10.0784 | shrubs | 18 m | Csa | 2002 - 2006 | - |
| 13 | IT-Ro1 | Roccarespampani1 | Italy | 42.4081, 11.9300 | deciduous broadleaf trees | 235 m | Csa | 2000 - 2008 | Rey et al. (2002) |
| 14 | IT-Ro2 | Roccarespampani2 | Italy | 42.3903, 11.9209 | deciduous broadleaf trees | 160 m | Csa | 2002 - 2008, 2010 - 2012 | Tedeschi et al. (2006) |
| 15 | IT-SRo | San Rossore | Italy | 43.7279, 10.2844 | evergreen needleleaf trees | 6 m | Csa | 1999 - 2010 | Chiesi et al. (2005) |
| 16 | IT-Tol | Tolfa | Italy | 42.1897, 11.9216 | evergreen broadleaf trees | 473 m | Csa | 2005 - 2006 | - |

*1 KGCC = Climate abbreviations follow the Koeppen-Geiger-Climate-Classification: Cfa - warm temperate fully humid with hot summer, Csa - warm temperate with dry, hot summer. *2 Note all years from which we used information (even we didn't use the year in total) are included in the table.

**Fig. 1.**

---

## Author Comment (AC2) · 7 Feb 2017

**General Answer to the Reviewer: We thank the anonymous reviewers for reading and reviewing our manuscript. We agree with both reviewers that our site selection could benefit from improvement and that we mainly concentrated on the vegetation types, not respecting the applied management or treatment of each site. We also agree that more information has to be provided concerning the selected sites. Correspondingly, we corrected our site selection by excluding sites where the management is difficult to reconcile with our statistical analysis. As given in table 1 (Tab.1, see below) we ended up having 16 sites left (instead of the original 23), representing four different vegetation types (evergreen needleleaf trees; evergreen broadleaf trees; deciduous**

broadleaf trees & shrubs) and three countries (Spain; France & Italy). In addition we included missing information such as the total elevation; climate information (KGCC); the number of years of observations included per site, as well as publications that are relevant to the sites' description (this column will be completed) in table 1. Based on this updated site selection, we will re-run all statistical analysis on the data.

**Reply to the comments of reviewer 2:**

We gratefully acknowledge the insight on the detailed site information. As mentioned above, reviewer 2 rightly criticizes the site selection, and our results may be flawed when mainly concentrating on the vegetation type of each site and ignoring its management. Correspondingly, we cleaned our data set as presented in the updated version of table 1 (see above) and added missing and important site information.

Regarding the concern that we have grouped all sites according to their vegetation, we decided to use, next to the 'site', also the 'vegetation type' as a random factor within our statistical analyses. Doing so will allow us to take site-dependent and also vegetation-dependent effects into account. Thank you for bringing this point forward.

We could not find any figure where the symbol legend is missing. As given as an example below (Fig. 4), for every figure the symbol legend (giving the different vegetation types) are presented in the first panel.

—————————————————————

Tab.1: Site description (NEW SITE SELECTION). Further site information is available at: https://fluxnet.ornl.gov/.

| Nr. | SITE ID | SITE NAME | COUNTRY | COORDINATES (Lat., Long.) | VEGETATION | ELEVATION | KGCC *1 | YEARS *2 | REFERENCES |
|---|---|---|---|---|---|---|---|---|---|
| 1 | ES-ES1 | El Saler | Spain | 39.3460, -0.3188 | evergreen needleleaf trees | 10 m | Csa | 1999 - 2006 | Sanz et al. (2004) |
| 2 | ES-LgS | Laguna Seca | Spain | 37.0979, -2.9658 | shrubs | 2267 m | Csa | 2007 - 2008 | - |
| 3 | ES-Llu | Llano de los Juanes | Spain | 36.9266, -2.7521 | shrubs | 1600 m | Csa | 2005 - 2011 | Serrano-Ortiz et al. (2007) |
| 4 | ES-Ln1 | Lanjaron-Non intervention | Spain | 36.9721, -3.4739 | shrubs | 2301 m | Csa | 2009 | - |
| 5 | FR-FBn | Font-Blanche | France | 43.2408, 5.6792 | evergreen needleleaf trees | 436 m | Csa | 2009 - 2011 | - |
| 6 | FR-Pue | Puechabon | France | 43.7414, 3.5958 | evergreen broadleaf trees | 270 m | Csa | 2001 - 2011 | Rambal et al. (2004) |
| 7 | IT-Bon | Bonis | Italy | 39.4778, 16.5347 | evergreen needleleaf trees | 1170 m | Csa | 2005 - 2009 | - |
| 8 | IT-CA3 | Castel d'Asso3 | Italy | 42.3772, 12.0222 | deciduous broadleaf trees | 197 m | Csa | 2012 | - |
| 9 | IT-Cpz | Castelporziano | Italy | 41.7052, 12.3761 | evergreen broadleaf trees | 68 m | Csa | 1997, 2000 - 2008 | Garbulsky et al. (2008) |
| 10 | IT-Lec | Lecceto | Italy | 43.3036, 11.2698 | evergreen broadleaf trees | 314 m | Cfa | 2005 - 2009 | Chiesi et al. (2011) |
| 11 | IT-Non | Nonantola | Italy | 44.6902, 11.0911 | deciduous broadleaf trees | 20 m | Cfa | 2001 - 2003, 2006 - 2008 | Reichstein et al. (2003) |
| 12 | IT-Pia | Island of Pianosa | Italy | 42.5839, 10.0784 | shrubs | 18 m | Csa | 2002 - 2006 | - |
| 13 | IT-Ro1 | Roccarespampani1 | Italy | 42.4081, 11.9300 | deciduous broadleaf trees | 235 m | Csa | 2000 - 2008 | Rey et al. (2002) |
| 14 | IT-Ro2 | Roccarespampani2 | Italy | 42.3903, 11.9209 | deciduous broadleaf trees | 160 m | Csa | 2002 - 2008, 2010 - 2012 | Tedeschi et al. (2006) |
| 15 | IT-SRo | San Rossore | Italy | 43.7279, 10.2844 | evergreen needleleaf trees | 6 m | Csa | 1999 - 2010 | Chiesi et al. (2005) |
| 16 | IT-Tol | Tolfa | Italy | 42.1897, 11.9216 | evergreen broadleaf trees | 473 m | Csa | 2005 - 2006 | - |

*1 KGCC = Climate abbreviations follow the Koeppen-Geiger-Climate-Classification: Cfa - warm temperate fully humid with hot summer, Csa - warm temperate with dry, hot summer. *2 Note all years from which we used information (even we didn't use the year in total) are included in the table.

**Fig. 1.**

[Figure]

**Fig. 2.** Figure 4